# OpenReview forum: "Memory Forgetting Adapter Sculpting for Selective Multimodal Large Language Model Unlearning"
_ICLR.cc/2026/Conference — ICLR 2026 Conference Withdrawn Submission_

### Official Review · Reviewer_TAAW · 2025-10-20

**Soundness:** 2
**Presentation:** 3
**Contribution:** 2
**Rating:** 2
**Confidence:** 4

**Summary:**

This paper propose a method, MEMORY FORGETTING ADAPTER SCULPTING,  for machine unlearning in multimodal large language model. The paper also introduces a new benchmark to evaluate the unlearning methods. The results show that the proposed method achieves comparable performance across different methods.

**Strengths:**

1. This paper proposes a new benchmark,  S-MLLMUn for machine unlearning in multimodal large language model.

2. The writing of this paper is easy to understand.

**Weaknesses:**

1. The paper lacks several related work such as [1] and [2], which makes this paper not that convincing.

2. In line 72 'Specifically, we first fine-tune the MLLM on privacy-sensitive data using refusal labels, obtaining a Memory Forgetting Adapter (MFA).' MFA seems to be a rather normal fine-tuning method and could not reflect the term memory. Moreover, 'this MFA will propagate forgetting effects to unrelated knowledge' is more confusing. Now that such component will affect unrelated knowledge, why still use it. Such sentence makes me think MFA could not solve any challenge proposed in the paper.

3. Where is the introduction of the small set for retaining anchor? Morever, why the small set knowledge must be preserved as mentioned in line 76? why the other knowledge does not need to be retained?





## Reference ##

[1]Liu, Zheyuan, et al. "Modality-aware neuron pruning for unlearning in multimodal large language models." arXiv preprint arXiv:2502.15910 (2025).

[2]Li, Jiaqi, et al. "Single image unlearning: Efficient machine unlearning in multimodal large language models." Advances in Neural Information Processing Systems 37 (2024): 35414-35453.

**Questions:**

Please see weaknesses.

---

### Official Review · Reviewer_y2Vk · 2025-10-30

**Soundness:** 3
**Presentation:** 2
**Contribution:** 2
**Rating:** 4
**Confidence:** 3

**Summary:**

The paper proposes the Sculpted Memory Forgetting Adapter (SMFA) for selective MLLM unlearning, aiming to erase privacy-sensitive knowledge while preserving the model's general image understanding ability. Specifically, it begins with training the Memory Forgetting Adapter (MFA) via refusal-label fine-tuning, followed by sculpting the MFA through a retaining anchor-guided masking mechanism to suppress overgeneralization. Additionally, the authors construct the S-MLLMUn Bench to jointly evaluate forgetting effectiveness and retention of visual capabilities.

**Strengths:**

(1)	The introduction of S-MLLMUn Bench, a novel benchmark incorporating synthetically generated personal profiles and ophthalmic medical images, establishes a more rigorous evaluation standard for selective MLLM unlearning.

(2)	The experimental results demonstrate the superiority of SMFA, including comparisons with strong baselines (Table 1), ablation studies (Table 2), and a case study (Figure 6). Furthermore, the analysis of the impact of different unlearning methods on ophthalmic images (Figure 5) highlights SMFA’s continued effectiveness in handling more challenging modality.

**Weaknesses:**

(1)	The masking mechanism in SMFA relies on intuitive criteria (directional conflict and relative magnitude). During this process, essential parameters might be inadvertently masked, raising concerns about the model’s convergence. Therefore, it is crucial to investigate how the proposed method ensures stable convergence under such aggressive parameter masking.

(2)	While S-MLLMUn Bench is well-constructed, all experiments are conducted on synthetic data. It is unclear how SMFA would perform on real-world privacy-sensitive datasets, such as MLLMU-Bench[1]. Furthermore, the S-MLLMUn Benchmark could be strengthened if it could cover real-world data, for example public celebrity profiles.

(3)	The analysis of the key hyperparameter k, which controls the masking aggressiveness in Eq. (7), is conducted exclusively under a low forget ratio of 5% (as shown in Figure 4). The sensitivity and optimal tuning of k under more challenging conditions, such as higher forget ratios (e.g., 10% or 15%) or on datasets with different characteristics remain unexplored. This implies a significant lack of guidance for effectively configuring this parameter in practice.

(4)	The implementation details of the proposed SMFA framework lack sufficient granularity. It is not clearly specified whether the retaining anchor are applied to all model parameters or a specific subset of layers (e.g., only the linear layers or attention mechanisms). This ambiguity hinders the reproducibility of the method and a precise understanding of its operational scope.

(5)	In Table 2, the Retain Set should be placed on the right side, with the Forget Set on the left side, so that the experimental results align with Table 1.

[1] Huo J, Yan Y, Zheng X, et al. Mmunlearner: Reformulating multimodal machine unlearning in the era of multimodal large language models[J]. arXiv preprint arXiv:2502.11051, 2025.

**Questions:**

Please see Weaknesses.

---

### Official Review · Reviewer_YFkb · 2025-11-03

**Soundness:** 3
**Presentation:** 3
**Contribution:** 3
**Rating:** 6
**Confidence:** 4

**Summary:**

This paper introduces S-MLLMUn Bench and a corresponding method (SMFA) for MLLM unlearning. The core goal is to remove sensitive knowledge while preserving the model's general image understanding capabilities. The benchmark is well-constructed, and the SMFA method presents an adapter-sculpting mechanism to mitigate over-generalization during unlearning.

**Strengths:**

- Well-Motivated Benchmark: The introduction of S-MLLMUn Bench is a key strength. Its dual structure—pairing sensitive "image memory" queries with general "image understanding" queries for the same image—establishes a rigorous and much-needed evaluation protocol for selective MLLM unlearning.

- Novel and Insightful Method: The proposed SMFA framework is conceptually elegant. The two-stage process of first learning a "forgetting adapter" (MFA) and then "sculpting" it with a "retaining anchor" (RA) is a clever approach to achieve precise, localized unlearning.

- Extensive Empirical Evaluation: The paper provides comprehensive experiments across multiple base models, forgetting ratios, and against several strong baselines. The ablation studies and case analyses are valuable for understanding the method's components and behavior.

**Weaknesses:**

1. **Forgetting of the Original Model:** The evaluation pipeline relies on a model fine-tuned on the S-MLLMUn Bench dataset. But it is possible that this initial fine-tuning itself causes catastrophic forgetting of the model's general capabilities. If this original model is already impaired, the subsequent evaluation of whether unlearning methods preserve "image understanding" is not confidential.

2. **Lack of Conceptual Clarity and Positioning**: The core of SMFA(MFA, RA) is highly related to the well-established paradigms of Model Editing and Task Arithmetic (e.g., TIES-Merging). The paper fails to discuss this connection. Is the primary innovation the application of this paradigm to MLLM unlearning, or is it the specific masking mechanism?

3. **Incomplete Related Work**: The related work section omits discussion of the Task Arithmetic literature, which is a direct precursor to the technical approach taken in this paper. This is a significant oversight that weakens the scholarly foundation of the work.

**Questions:**

Could the authors provide evidence comparing their fine-tuned original model against the pre-trained MLLM (e.g., base LLaVA) on the image understanding evaluation set to prove no significant performance drop occurred during the initial memorization phase?

More discussion about task arithmetic methods in related work and experiment is encouraged.

---

### Note · Authors · 2025-11-13

I have read and agree with the venue's withdrawal policy on behalf of myself and my co-authors.